# Cellulose Acetate-g-Polycaprolactone Copolymerization Using Diisocyanate Intermediates and the Effect of Polymer Chain Length on Surface, Thermal, and Antibacterial Properties

**DOI:** 10.3390/molecules27041408

**Published:** 2022-02-19

**Authors:** Abdessamade Benahmed, Khalil Azzaoui, Abderahmane El Idrissi, Hammouti Belkheir, Said Omar Said Hassane, Rachid Touzani, Larbi Rhazi

**Affiliations:** 1Laboratory of Applied Chemistry and Environment (LCAE), Faculty of Sciences, University Mohammed Premier, PB 4808, Oujda 60046, Morocco; samadben44@gmail.com (A.B.); ab.elidrissi@yahoo.fr (A.E.I.); hammoutib@gmail.com (H.B.); r.touzani@ump.ac.ma (R.T.); 2Centre de Recherche, Ecole des Hautes Etudes d’Ingénierie EHEIO, Oujda 60046, Morocco; 3Département de Physique Chimie Faculté des Sciences et Techniques, Université des Comores, BP 2585, Moroni 99999, Comoros; said_omar2000@yahoo.fr or; 4Institut Polytechnique UniLaSalle Transformations & Agro-ResourcesResearch Unit (ULR7519) 19 rue Pierre Waguet, BP 30313, 60026 Beauvais, France

**Keywords:** biodegradable polymers, polycaprolactone, polysaccharides, cellulose acetate, grafting to, molecular characteristics, food packaging, biomedical applications

## Abstract

The need for biodegradable and biocompatible polymers is growing quickly, particularly in the biomedical and environmental industries. Cellulose acetate, a natural polysaccharide, can be taken from plants and modified with polycaprolactone to improve its characteristics for a number of uses, including biomedical applications and food packaging. Cellulose acetate-g-polycaprolactone was prepared by a three-step reaction: First, polymerization of ε-caprolactone via ring-opening polymerization (ROP) reaction using 2-hydroxyethyl methacrylate (HEMA) and functionalization of polycaprolactone(PCL) by introducing NCO on the hydroxyl end of the HEMA-PCL using hexamethyl lenediisocyanate(HDI) were carried out. Then, the NCO–HEMA-PCL was grafted onto cellulose acetate (using the “grafting to” method). The polycaprolactone grafted cellulose acetate was confirmed by FTIR, the thermal characteristics of the copolymers were investigated by DSC and TGA, and the hydrophobicity was analyzed via water CA measurement. Introducing NCO-PCL to cellulose acetate increased the thermal stability. The contact angle of the unreacted PCL was higher than that of cellulose acetate-g-PCL, and it increased when the chain length increased. The CA-g-PCL50, CA-g-PCL100, and CA-g-PCL200 showed very high inhibition zones for all three bacteria tested (*E. coli*, *S. aureus*, and *P. aeruginosa*).

## 1. Introduction

The demand for degradable and biocompatible polymers is rapidly increasing, especially in the biomedical and environmental sectors. These materials are available in large quantities, and they possess numerous advantageous properties [1]. However, natural polymers, mainly polysaccharides, consist of large, rigid molecules and cannot be processed very easily with the usual processing technologies of thermoplastic polymers [2]. As a consequence, many attempts are made to modify them, both to improve processability and to adjust their properties for the intended application. Natural polymers can be modified physically by plasticization, or chemically through the reaction of their active –OH groups with other chemical functional groups [3,4,5]; benzylation of wood, plasticization of starch [6], and grafting of cellulose [7] or cellulose acetate with aliphatic polyesters are typical examples of such modification [8,9,10,11,12,13,14,15].

Polysaccharides are a group of carbohydrates with a large polymeric oligosaccharide formed through glycosidic bonds in the presence of multiple monosaccharides [16]. In nature, the main sources of polysaccharides (e.g., pectin, cellulose, starch) are from animals (chitosan, chitin, glycosaminoglycan), the microbial domain (e.g., dextran, pullulan, xanthan gum, gellan gum), and algal origin (e.g., alginate and carrageenan). Depending on the composition of the monosaccharide units, polysaccharides can be classified as homopolymers(consisting of monosaccharide repeats such as glycogen, starch, cellulose, pullulan, and pectin) or heteropolymers(composed of different monosaccharide units, such as chitosan, heparin, hyaluronic acid, chondroitin sulfate, creatine sulfate, heparan sulfate, and dermatan sulfate) [17,18].Due to their abundance, lower analytic cost, biocompatibility, biodegradability, non-toxicity, water solubility, and viability, polysaccharides are considered as some of the most suitable biomaterials in nanomedicine [19]. In addition, polysaccharides have many reactive functional groups in their core chemical structure, mainly hydroxyl, amino, and carboxylic acid groups, which facilitate the extraction process and contribute to their structural and reactional diversity. Based on the biodegradability and non-toxicity of the final products obtained, many investigations are currently focused on polysaccharides and their families for valorization as nanoparticles (such as nanogels or micelles), especially in drug delivery systems [20].

Polysaccharides grafted with synthetic polymers have attracted considerable attention from researchers in recent decades. These polymers are highly popular as scaffold materials, as they have defined chemistry and easy processing and tailoring ability, and can be modified to achieve desired properties for specific applications. Other merits include cost efficacy, uniform large-quantity production, and longer shelf life. In addition, their physicochemical and mechanical properties, such as tensile strength, elastic modulus, and degradation rate, are comparable to bone [21]. However, these polymers are not bioactive, and hence they can elicit inflammatory responses inside the host. Polymers such as polylactic acid (PLA), polyglycolic acid (PGA), poly-l-lactide (PLLA), poly ε-caprolactone (PCL), polylactic–glycolic acid (PLGA) copolymers, and polyhydroxy–alkanoates (PHA) are classified as biodegradable synthetic polymers. Within the class of synthetic materials, PCL has recently drawn much attention for biomedical applications, including bone tissue engineering [21].

Polycaprolactone (PCL) is a biodegradable polymer with excellent mechanical qualities that can be utilized in food packaging applications. However, most polymeric materials used in food packaging are not biodegradable. The primary focus of non-biodegradable polymer research is on turning them into biodegradable products by using additives. PCL is an additive that is biodegradable polyester, and is compatible with most materials [22]. It is an aliphatic semi-crystalline polymer with a melting temperature ranging from 59–64 °C (i.e., above body temperature) and a glass transition temperature of −60 °C. Hence, at physiological temperature, the semi-crystalline PCL attains a rubbery state, resulting in high toughness and superior mechanical properties (high strength and elasticity depending on its molecular weight). It is non-toxic and tissue compatible, and hence is widely used inresorbable sutures, in scaffolds for regenerative therapy, and in drug delivery applications. PCL exhibits a longer degradation time (2–3 years) and is degraded by microorganisms or by hydrolysis of its aliphatic ester linkage under physiological conditions. Due to the presence of five hydrophobic –CH2 moieties in its repeating units, PCL degrades the slowest among certain polyesters. The erosion rate of nanofiber matrices made from polyesters follow the order of PGA > PLGA >PLLA > PCL [21].

Three methods of grafting polymer chains on the surface of polysaccharides have been reported in the literature (Figure 1). The first is the “grafting to” method, in which the end-functionalized polymers react with the polysaccharide backbone. The second is the “grafting from” method, in which polymer chains are grown from the polysaccharide backbone. The third is the “grafting through” method, which involves (co)polymerization of macromonomers.

The second method is currently employed the most because it allows higher grafting densities to be reached due to the lower steric hindrance of diffusing small monomers compared to polymer chains, compared with the grafting to approach, in which the already grafted polymer chains may shield reactive sites on the surface [23,24,25]. In the majority of grafting studies, grafting from is reported to provide higher grafting densities [26,27,28,29], although examples have been reported where grafting to has provided similar or even higher grafting densities. In the grafting from approach, reactive sites along the main chain can be simply created by chemical treatment or irradiation, followed by the addition of monomer to generate graft copolymer. The grafting onto approach generally has low reaction efficiency due to the low activity of macromolecular reactions [30,31].

In our laboratory, the development of a composite based on polylactic acid (PLA) and treated with hydroxyapatite (HApt) usingpoly-caprolactone(PCL) as an adjuvant significantly improved the resistance and rigidity of the composite, which can extend the physico-chemical and biological applications of this material [32].

In this study, cellulose acetate-g-PCL copolymers were prepared by using diisocyanated intermediates of PCL (Figure 2). A similar approach was already reported in the literature regarding the use of an NCO function, such as papers by Paquet et al. concerning the surface modification of cellulose by PCL grafts [33] and Tabaght et al. [34] concerning the synthesis of novel biodegradable cellulose derivatives using the grafting onto process and 1,6-hexamethylene diisocyanate as a coupling agent. The cellulose used in this study was extracted from esparto *Stipatenacissima* of Eastern Morocco following the procedure developed by El Idrissi et al. [35] to prepare cellulose acetate by the method described by El Berkany et al. [36]. To our knowledge, there is no published paper reporting the grafting of cellulose acetate with polycaprolactone by using diisocyanated intermediates. The main idea in this research was the polymerization of ε-caprolactone via the ring-opening polymerization (ROP) reaction using 2-hydroxyethyl methacrylate to create a high-reactivity double bond at the extremity of the polymer (HEMA-PCL), and to find a correlation between structure (chain length) and some properties of the samples (thermal properties, antibacterial activity, hydrophobicity, etc.).

## 2. Results and Discussion

### 2.1. Synthesis of Polycaprolactone (PCL)

Polycaprolactone (HEMA-PCL) was prepared by reaction between the monomer ε-caprolactone and 2-hydroxyethyl methacrylate via ring-opening polymerization (ROP) reaction (Figure 3). The chemical structures of HEMA-PCL were confirmed by 1H NMR and FTIR spectroscopy.

Figure 4 shows the 1H NMR spectrum of HEMA-PCL. The peak at 5.33 ppm (peak a) is attributed to the proton peak of the active double bonds; the peaks at 4.28 ppm (peaks c,d) are attributed to the methylene group of O=COCH2CH2O-; the peak at 4.02 (peak i) is attributed to the internal methylene proton of PCL (2H, –CO(CH2)4- CH2O–); the peak at 3.6 ppm (peak i’) is attributed to the hydroxyl peak at the end of the polymer chain; and the peak at 2.6 ppm (peak b) is attributed to the methyl group of (-CH3)C=CH2. Peaks for the polymer chain (PCL) were observed at 2.29 ppm (peaks e, e’), 1.62 ppm (peaks f,f’,h,h’), and 1.38 ppm (peaks g,g’). The figure indicates that the synthesis of HEMA-PCL macromonomers was successful.

In the FTIR spectrum of HEMA-PCL (Figure 5), the peaks at 1728 and 3548 cm^−1^ are the stretching vibrations of C=O and -OH, the peak at 1293 cm^−1^ is the C-O stretching vibration, the peak at 1247 cm^−1^ is the C-O-C stretching vibration, the peak at 1635 cm^−1^ is the C=C stretching vibration, and the peak at 2946 ${\mathrm{cm}}^{-1}$ is the bending and stretching vibration of the C-H of methyl and methylene groups. These peaks confirmed the correct structure of the HEMA-PCL polymer [37].

### 2.2. Determination of DP (for PCL) by Viscosity Measurements

The flow time measurements of the concentrations varying from 1 to 2 g/dL were carried out at a temperature of 30 °C using a Cannon–Ubbelohde type tube. Dimethyl formamide (DMF) was used as solvent.

It is critical to work with solutions that have the appropriate concentration and no impurities. The solutions must not be too diluted (loss of precision) or too concentrated (interchain interactions). The ideal concentration range for viscosity measurements in a good solvent corresponds to: 1.1≤ $\eta_{r}$≤1.4 [38]. Before each measurement, the capillary tube is washed with acetone, and then with DMF. Manual measurement of the flow time (> 100 s) is sufficient to determine the viscosity. However, great care must be taken with temperature control.

The flow time of the pure solvent,$t_{0}$, and of different PCL concentrations, t, was measured. Based on these time values, the relative viscosity $\eta_{r}$ and the specific viscosity $\eta_{\mathrm{sp}}$ were calculated using Equations (1) and (2) [38]:(1)$$\eta_{r}=\frac{\eta }{\eta_{0}}=\frac{t}{t_{0}}$$
(2)$$\eta_{\mathrm{sp}}=\frac{\eta -\eta_{0}}{\eta_{0}}={\;\eta }_{r}-1$$

We used Huggins and Kraemer analyses to determine the intrinsic viscosity, [η], of PCL samples in a DMF solvent. In Huggins’ method, intrinsic viscosity [η] is defined as the ratio of the increase in relative viscosity ($\eta_{\mathrm{sp}}$) to concentration (C) when the latter tends towards zero [39].
(3)$$\eta_{\mathrm{red}}=\frac{\eta_{\mathrm{sp}}}{C}=k^{\prime }{\left[\eta \right]}^{2}C\;+\left[\eta \right]$$
where K′ is Huggins constant.

The Kraemer method uses a similar approach, except the Kraemer equation is defined in terms of the natural logarithm of the relative viscosity [40]:(4)$$\eta_{\mathrm{inh}}=\frac{\mathrm{Ln}\left(\eta_{r}\right)}{C\;}=k^{\prime \prime }{\left[\eta \right]}^{2}C\;+\left[\eta \right]$$
where K″ is the Kraemer constant.

The measurement results are presented in Table 1, Table 2 and Table 3. We determined the intrinsic viscosity of PCL by averaging the y-intercept values for linear fits of the Kraemer and Huggins plots, which are shown in Figure 6, Figure 7 and Figure 8.

Molar mass and viscosity could be related in an empirical manner through the Mark–Houwink–Sakurada (M–H–S) equation: [38]
(5)$$\left[\eta \right]={\mathrm{kM}}_{v}{}^{a}$$
where [η] is the intrinsic viscosity, K and a are constants varying with the polymer, solvent, and temperature under consideration, and M is usually one of the relative molecular mass averages.

The intrinsic viscosity [η] according to Naar et al. equation is: [41]
(6)$$\left[\eta \right]=\left(\frac{1}{C}\right).\sqrt{\left(2\left(\eta_{\mathrm{sp}}-\mathrm{Ln}\left(\;\eta r\;\right)\right)\right)}$$

The results of the calculations of intrinsic viscosity with the Mark–Houwink–Sakurada equation and the equation of Naar et al. [41] are shown in Table 4.

Applying the Mark–Houwink equation [42], for the system (PCL/DMF) at T= 30 °C, the viscometric molecular weight was calculated for the three polymers, and the results are presented in Table 5. The viscometric molecular weight is transformed to the average molecular weight using the equation:(7)$${\left[\frac{\left(\mathrm{DP}\right)v}{\left(\mathrm{DP}\right)n}\right]}^{a}=\left[\frac{Г\left(3+a\right)}{2^{\left(1+a\right)}}\right]$$
where a is the Mark–Houwink coefficient and Г is the gamma function.

For the system (PCL/DMF), k and a parameters are defined in the literature [43] at T = 30 °C: k = 1.91. 10^−4^ and a = 0.73.

Mathematically, we find that:(8)Г(3+a)= Г(3+0.73)=Г(3.73)=4.32

Therefore:(9)$${\left[\frac{\left(\mathrm{DP}\right)v}{\left(\mathrm{DP}\right)n}\right]}^{0.73}=\left[\frac{4.32}{2^{\left(1.73\right)}}\right]\;\Leftrightarrow \left(\mathrm{DP}\right)n=\frac{\left(\mathrm{DP}\right)v}{{\left(\frac{4.32}{2^{1.73}}\right)}^{\frac{1}{0.73}}}$$

### 2.3. Measurement of Hydroxyl Value

Hydroxyl values of HEMA-PCLs were measured according to AOAC Official Method [44]. In order to verify the reliability of this method, the hydroxyl value $I_{oh}$ of polyethylene glycol 2000 was measured, with results as follows:(10)$$I_{oh}=\frac{\left(\mathrm{volume}\;\mathrm{of}\;\mathrm{blank}\;\left(\mathrm{mL}\right)-\;\mathrm{volume}\;\mathrm{of}\;\mathrm{sample}\;\left(\mathrm{mL}\right)\right)\times N\left(\mathrm{NaOH}\right)\times 56.1}{g\left(\mathrm{sample}\right)}$$
(11)$$I_{oh}=\frac{\left(47.2\left(\mathrm{mL}\right)-40.6\left(\mathrm{mL}\right)\right)\times 0.5\times 56.1}{3\;g}=61\;\mathrm{mg}/g$$

From the definition of the hydroxyl value, which is the amount of KOH (in milligrams) needed to neutralize the acetic acid formed from the acetylation of 1 g of sample, the following relation is deduced: [44]
(12)$$\mathrm{Mn}=\frac{56.1\times 1000\times \mathrm{Fn}}{I0\;H}=\frac{56.1\times 1000\times 2}{61.7}=1818.5\;g/\mathrm{mol}\approx 2000\;g/\mathrm{mol}$$
where Fn is the number of OH functions.

From the results obtained, it can be seen that this method yields acceptable results for polymers of relatively low molar mass.

The results of the hydroxyl value measurement for the three samples are shown in Table 6.

From the tables, it can be seen that the results obtained by the viscosity measurement of molar mass and by the titration of hydroxyl value (although some values seem to be different) are in agreement with the theoretical results and those found in the literature [45], i.e., that the degree of polymerization increases as the concentration of the initiator (HEMA) decreases. Figure 9 shows the evolution of the degree of polymerization as a function of the amount of moles of initiator (HEMA).

### 2.4. Synthesis of HEMA-PCL-NCO

The three HEMA-PCL-NCO samples were prepared by bulk reaction between hydroxyl-terminated HEMA-PCL and HDI, and the reaction is shown in Figure 10.

By introducing NCO groups onto the HEMA-PCL terminals, new sharp peaks at 2270 ${\mathrm{cm}}^{-1}$ representing the NCO groups and 1520 ${\mathrm{cm}}^{-1}$ representing the newly formed amide —NH groups were shown (Figure 11).

The FTIR spectrum of HEMA-PCL-NCO prepared with HDI showed a slightly increased peak, around 780 ${\mathrm{cm}}^{-1}$ (marked by an arrow), indicating the presence of the methylene group originating from HDI (Figure 11).

### 2.5. Preparation of CA-g-PCL

The CA-g-PCL copolymer was prepared by reacting HEMA-PCL-NCO with cellulose acetate. The reaction of copolymerization is shown in Figure 12.

The FTIR spectra of CA-g-PCL copolymers are shown in Figure 11. With grafting of NCO–PCL onto cellulose acetate, the spectra showed new peaks around 1730 ${\mathrm{cm}}^{-1}$, indicating the amide ester linkage, and around 1540 ${\mathrm{cm}}^{-1}$, for the amide –NH group. At the same time, the peak around 3400 ${\mathrm{cm}}^{-1}$ and around 2270 ${\mathrm{cm}}^{-1}$ corresponding to hydroxyl (OH) and NCO groups respectively disappeared in the CA-g-PCL spectrum; this is due to the interaction between the two functional groups as shown in Figure 12.

### 2.6. Effect of Degree of Polymerization (for PCL) in Grafting Density

The percent grafting and product yield of CA-g-PCL copolymers are shown in Table 7. The percent grafting was calculated according to the relation:(13)$$\mathrm{Grafting}\left(\%\right)=\frac{\left(\mathrm{weight}\;\mathrm{of}\;\mathrm{CA}\;g\;\mathrm{PCL}\;\mathrm{copolymer}\right)–\left(\;\mathrm{weight}\;\mathrm{of}\;\mathrm{CA}\right)}{\left(\mathrm{weight}\;\mathrm{of}\;\mathrm{CA}\right)}$$

The yield was calculated using the relation:(14)$$\mathrm{Yield}\left(\%\right)=\frac{\left(\mathrm{weight}\;\mathrm{of}\;\mathrm{CA}\;g\;\mathrm{PCL}\;\mathrm{copolymer}\right)}{\left(\mathrm{weight}\;\mathrm{of}\;\mathrm{CA}\right)+\left(\mathrm{weight}\;\mathrm{of}\;\mathrm{HEMA}\;\mathrm{PCL}\;\mathrm{NCO}\right)}$$

The analysis of results in Table 7 shows that the percent of grafting decreased with increasing DP, and this is logical because the already grafted polymer chains can shield the reactive sites on the surface.

The yield was relatively constantly over 88%, indicating that the HEMA-PCL-NCO added to the mixture was effectively incorporated into the CA.

### 2.7. Thermal Properties of CA-g-PCL

The thermal transition behavior of CA-g-PCL copolymers measured by TGA and DSC is shown in Figure 13 and Figure 14.

As shown in Figure 13, the degradation temperature (Td) was found to be around 290 °C for cellulose acetate and around 315 °C for PCL, but The Td of the copolymer (CA-g-PCL) at the point where the degradation started was significantly higher (340 °C). By grafting PCL onto cellulose acetate, the thermal stability was largely increased.

As shown in Figure 14, the melting point of the PCL matrix was about 64 °C, and the addition of cellulose acetate with different DPs of PCL did not affect the melting of PCL. The obtained results are in accordance with previous studies with regard to the effects of modified or unmodified cellulose from different sources on the melting of some polymer matrices, such as plasticized starch [46], poly(ethylene oxide) [47], and PCL [48,49].

### 2.8. Effect of DP of PCL on the Contact Angle

The contact angles (CAs) of the surface of CA-g-PCL were measured to determine the effect of PCL DP on the hydrophobicity of the following samples: PCL, CA-g-PCL50, CA-g-PCL100, and CA-g-PCL200.The results showed that the angles were 86.9°, 58.2°, 59°, and 62.5°, respectively (Figure 15).

The cellulose acetate decreased the surface contact angle because it is hydrophilic, but by increasing the chain length of PCL, the contact angle increased because the PCL is hydrophobic.

### 2.9. SEM Analysis of Surfaces

In order to study the morphology of CA-g-PCL, scanning electron microscopy (SEM) analysis was performed. Figure 16 shows the morphology of the prepared samples.

The SEM image showed that the surface of cellulose Acetate (CA) changed, as illustrated in Figure 16. Thus, the CA surface seemed to become smoother after grafting with PCL, and this is consistent with the results of the contact angles. Moreover, few spherical particles were detected at the surface. From these micrographs, one can deduce that the modification is not homogeneously distributed at the CA surface.

### 2.10. Impact of Chain Length on Antibacterial Activity

The antibacterial activity of CA-g-PCL50, CA-g-PCL100, and CA-g-PCL200 were tested against three pathogenic bacteria, *E. coli, P. aeruginosa*, and *S. aureus*, using the solid medium disk diffusion method. This involved soaking paper discs with a volume of 20 µL of the colloidal solution of the three copolymers.

Two negative and positive antibiotic discs were used as controls. A disc was impregnated with DMSO solvent used for the dispersion of powders. The prepared wells were incubated with moderate agitation at 37 °C for 24 h. The test was carried out twice at the rate of three wells/sample each time.

*E. coli, S. aureus*, and *P. aeruginosa* were chosen because they are the most common bacteria found in bone infections, such as osteomyelitis. The antibacterial activity of the prepared samples against the three bacteria is shown in Figure 17 and Figure 18. The diameters of the inhibition zones of samples against the three bacteria are given in Table 8.

CA-g-PCL50, CA-g-PCL100, and CA-g-PCL200 showed a large high inhibition zone for all three bacteria (*E. coli, S. aureus*, and *P. aeruginosa*). This antibacterial activity may be due to the presence of biomolecules in the samples [50], which would inhibit the enzymes necessary for bacterial growth and replication, as well as the presence of polymers with low solubility in water, which would limit the availability and possibility of penetration of the polymer into the cell wall [51]; this is why the antibacterial activity decreased when the chain length of PCL increased because the copolymer is more hydrophilic. In that regard, the CA-g-PCL50 copolymer showed higher antibacterial activity against both Gram-negative and Gram-positive bacteria than CA-g-PCL100 or CA-g-PCL200.

## 3. Materials and Methods

### 3.1. Materials

Cellulose acetate was synthesized at our laboratory. The materials ε-caprolactone, tin octoate (SnOct2), tetrahydrofuran (THF), hexamethylene diisocyanate (HDI 98%), methanol (MeOH), dichloromethane (DCM), dimethyl sulfoxide (DMSO), toluene, dibutyltin dilaurate (DBTDL), and dimethylformamide (DMF) were purchased from Sigma-Aldrich (St. Louis, MS, USA).

### 3.2. Synthesis of Polycaprolactone (PCL)

The monomer ε-caprolactone (430 mmol) was introduced into a round-bottomed flask equipped with a magnetic stirrer. Then, different volumes of 2-hydroxyethyl methacrylate were added to the reaction mixture (depending on the desired DP of PCL), and a catalytic amount of Sn(Oct)_2_ (2 wt% of monomer) was added to the reaction flask. The reaction was carried out under magnetic stirring and argon flow for 3 h at 130 °C. The mixture was dissolved in a minimum of dichloromethane, then precipitated with methanol and dried in a vacuum oven to a constant weight.

### 3.3. Synthesis of DI-Terminated PCL (PCL-NCO)

HEMA-PCL-NCO was prepared by reacting PCL with HDI in bulk (Figure 10). PCL (10 g), HDI (4.5 g), and DBTDL as catalyst (1 wt%) were charged in a 3-necked flask equipped with a mechanical stirrer, reflux condenser, and N2 gas atmosphere. The mixture was then held at 80 °C for 3 h. The unreacted HDI was removed by a rotary vacuum evaporator.

### 3.4. Synthesis of Cellulose Acetate-g-PCL

Cellulose acetate (1 g) was dissolved in anhydrous THF (50 mL), and separately, PCL-NCO (7.5 g) was dissolved in anhydrous THF (100 mL) and an appropriate amount of DBTDL. The cellulose acetate solution was slowly added to the PCL-NCO solution while stirring for 30 min. The mixture was then stirred at 80 °C for 3 h under N2 gas atmosphere. The reaction mixture was cooled to room temperature, and then excess methanol was added to the mixture. The precipitate was recovered by vacuum filtration and the unreacted PCL-NCO was removed by Soxhlet extraction with toluene for 24 h.

### 3.5. Instruments/Equipment

#### 3.5.1. Fourier Transform Infrared Spectroscopy (FTIR)

To evaluate the grafting of PCL onto cellulose acetate, ATR-FTIR was used. The spectra were determined on a FT/IR-4700 spectrometer (JASCO, Japan). The data were collected in the spectral range of 400 to 4000 ${\mathrm{cm}}^{-1}$. Each spectrum was obtained by averaging 32 scans at a resolution of 4 ${\mathrm{cm}}^{-1}$. Background was acquired before each analysis.

#### 3.5.2. Differential Scanning Calorimetry (DSC)

Differential scanning calorimetry (DSC)analysis, which enables determination of the latent heat of molecules during a phase transition, was performed with a TA DSC Q20 device (TA Instruments, New Castle, DE, USA). About 10 mg of each sample was placed in sealed aluminum capsules and subjected to 2 scans from −40 to 200 °C with a rate of 10 °C/min under inert atmosphere.

#### 3.5.3. Standard Thermogravimetric Analysis (TGA)

Standard thermogravimetric analysis (TGA)was performed using a TGAQ500 device (TA instruments, New Castle, DE, USA) at a heating rate of 20 °C min^−1^ under 50 mL min^−1^ nitrogen flow within the range of 25 to 600 °C. The sample weight was between 8 and 12 mg.

#### 3.5.4. Viscosity Measurements

The intrinsic viscosity was measured using capillary Cannon-Ubbelohde viscometer tube (Normalab France S.A.S., Valliquerville, France) (Figure 19). Dimethyl formamide (DMF)was used as solvent andthe characteristic Mark–Houwinkconstants of the couple (DMF/PCL) at T = 30 °C were k = 1.94 and a = 0.73.

To prepare the stock solution, 2 g of the synthesized product was dissolved in 100 mL of DMF in an Erlenmeyer flask under stirring. Then, the solutions were prepared with concentrations varying from 1 to 2 g/dL, and the volume of each solution was 25 mL.

#### 3.5.5. Measurement of Hydroxyl Value

The hydroxyl value of the PCL was determined by alkaline titration after acetylation of the terminal hydroxyl groups. PCL and acetic anhydride/pyridine solution (5% *w*/*w*, 200 mL) were mixed and reacted at 100 °C for 3 h under N_2_. After being cooled to room temperature, the obtained mixturewas titrated with 0.5 NNaOH solution until neutralization.

The hydroxyl value of PCL was calculated with the following equation:Hydroxyl value = [(A−B)× N]/W
where A is the titration volume (mL) of the NaOH solution for the blank, Bis the titration volume (mL) of the NaOH solution for PCL, N is the normality of the NaOH solution, and W is the dry weight (g) of PCL.

#### 3.5.6. Water Contact Angle (WCA)Tests

The contact angles of the PCL and copolymers were measured with an optical contact angle goniometer (CAM 100, KSV Instruments Ltd., Helsinki, Finland). This compact video-based instrument measures contact angles between 1° and 180° with an accuracy of ±1°. The computer software provided by KSV Instruments Ltd. precisely recorded and measured the contact angles and took pictures of the measured contact angle values. Three samples for each group were tested to calculate the average.

#### 3.5.7. Scanning Electron Microscopy (SEM)

Scanning electron microscopy was carried out with HiroxTabletop SEM SH-5500P (HIROX, Japan) with an acceleration voltage of 10 kvto 15 kvin high vacuum.

## 4. Conclusions

In the present work, CA-g-PCL copolymers were prepared by using di-isocyanated intermediates for more reactivity.

A graftingto approach was utilized to covalently graft the cellulose acetate with PCL to different target DPs (CA-g-PCL). The characterization of ungrafted PCL and PCL-grafted cellulose acetate showed an increase in thermal stability (290 °C versus 340 °C), and the addition of cellulose acetate with different DPs of PCL did not affect the melting point of PCL(about 64 °C).

The contact angle decreased from 86.9° for PCL to 58.2° for CA-g-PCL50 by the addition of cellulose acetate because it is hydrophilic, but by increasing the chain length of PCL, the contact angle also increased because PCL is hydrophobic.

Therefore, the CA-g-PCL50 copolymer showed higher antibacterial activity against both Gram-negative and Gram-positive bacteria (*E. coli, S.aureus*, and *P.aeruginosa*) compared withCA-g-PCL100 and CA-g-PCL200, implying that when the chain length of PCL increases, the antibacterial activity decreases because the copolymer becomes more hydrophilic.

## Figures and Tables

**Figure 1 molecules-27-01408-f001:**
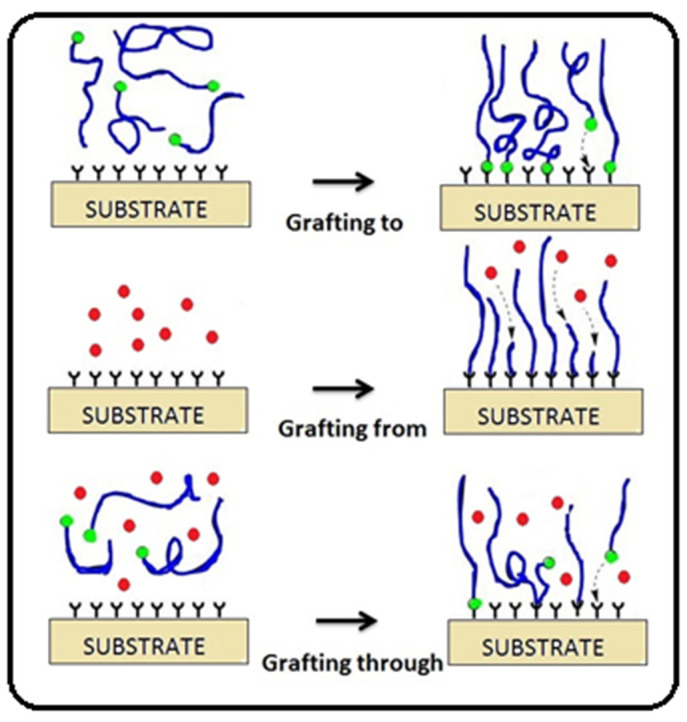
Schematic illustration of grafting to, grafting from, and grafting through approaches.

**Figure 2 molecules-27-01408-f002:**
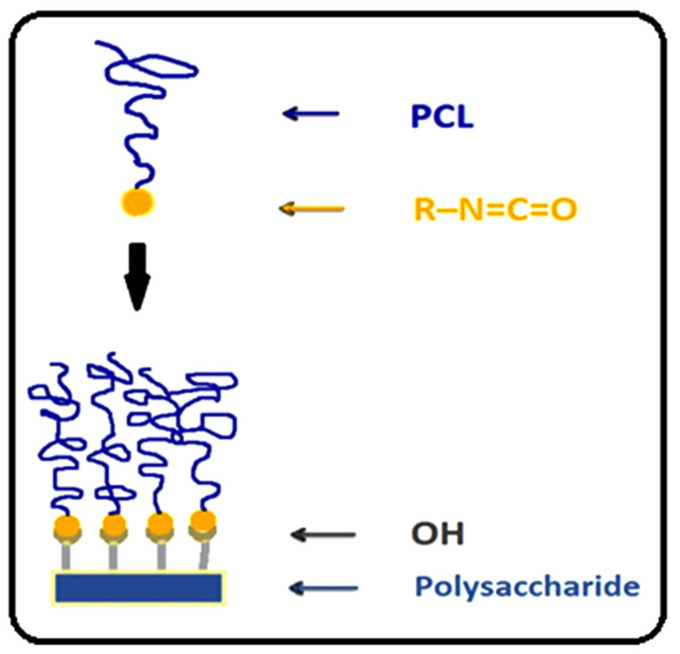
Schematic illustration of grafting PCL to cellulose acetate using diisocyanated intermediates.

**Figure 3 molecules-27-01408-f003:**
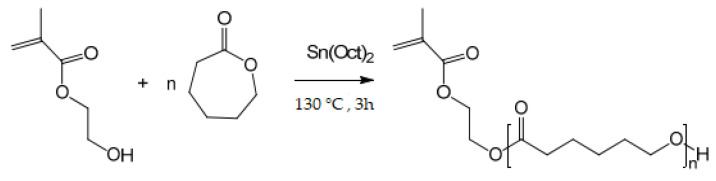
Synthesis of HEMA-PCL macromonomer via ring-opening polymerization (ROP) reaction.

**Figure 4 molecules-27-01408-f004:**
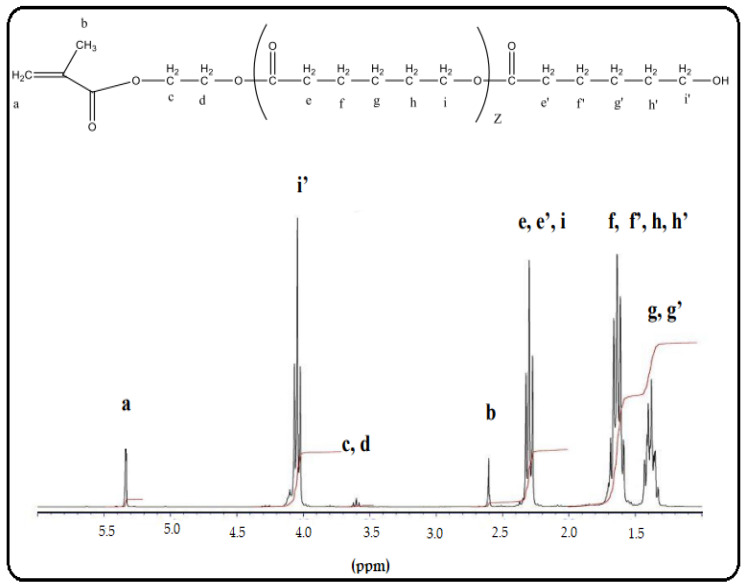
Proton nuclear magnetic resonance (1H NMR) spectrum of HEMA-PCL. Letters were attributed to the proton peaks.

**Figure 5 molecules-27-01408-f005:**
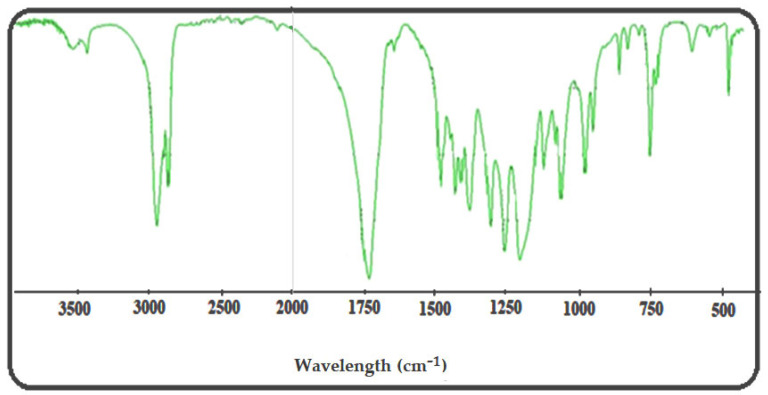
FTIR spectrum of HEMA-PCL.

**Figure 6 molecules-27-01408-f006:**
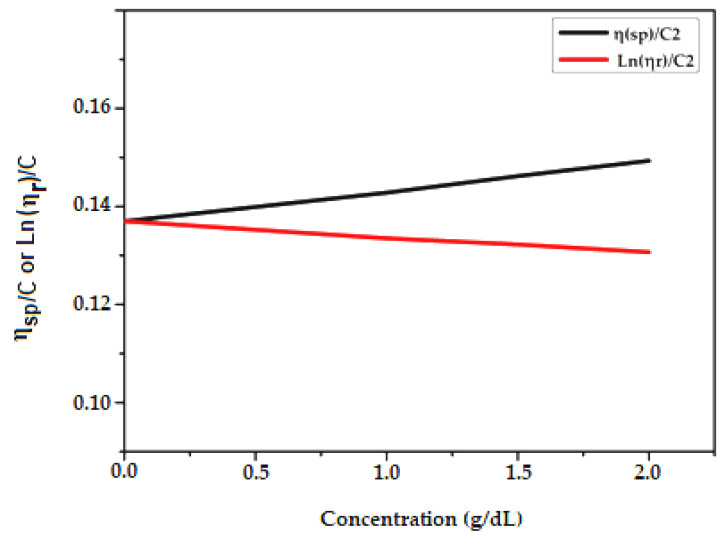
Zero concentration extrapolation of variation of $\eta_{\mathrm{sp}}$/C (according to Huggins) and Ln($\eta_{r}$ )/C(according to Kraemer) as a function of HEMA-PCL50 concentration in g/dL.

**Figure 7 molecules-27-01408-f007:**
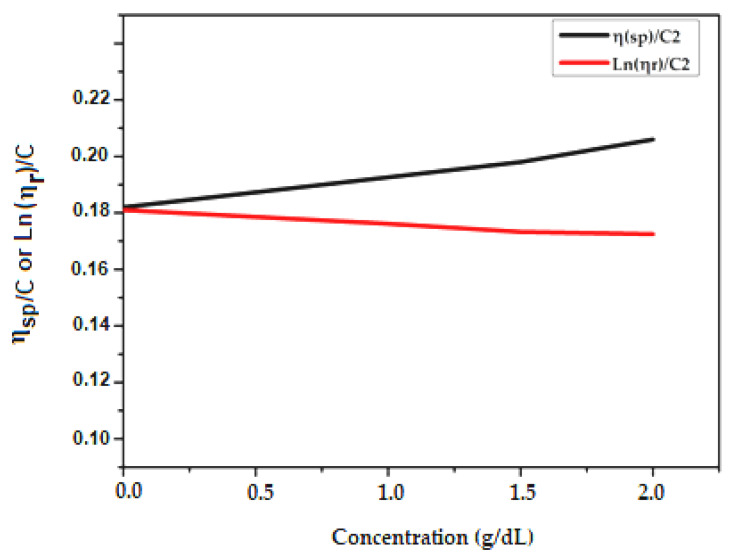
Zero concentration extrapolation of variation of $\eta_{\mathrm{sp}}$/C (according to Huggins) and Ln($\eta_{r}$ )/C (according to Kraemer) as a function of concentration of HEMA-PCL100in g/dL.

**Figure 8 molecules-27-01408-f008:**
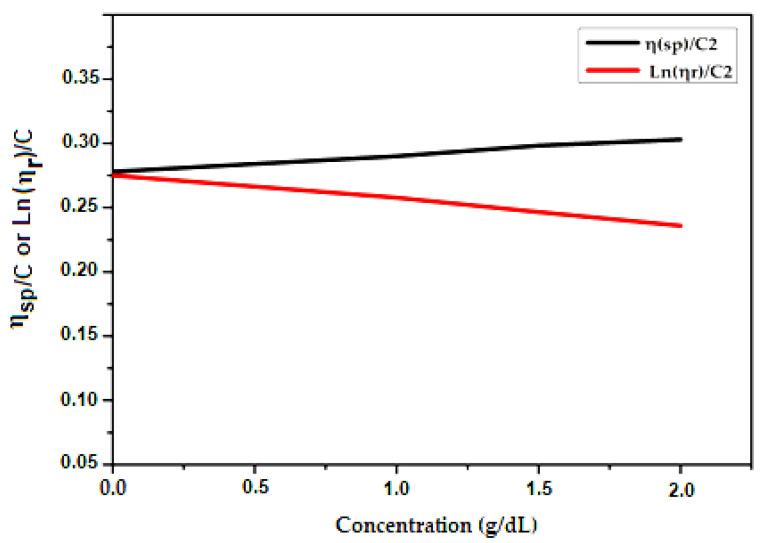
Zero concentration extrapolation of variation of $\eta_{\mathrm{sp}}$/C (according to Huggins) and Ln($\eta_{r}$ )/C (according to Kraemer) as a function of concentration of HEMA-PCL200 in g/dL.

**Figure 9 molecules-27-01408-f009:**
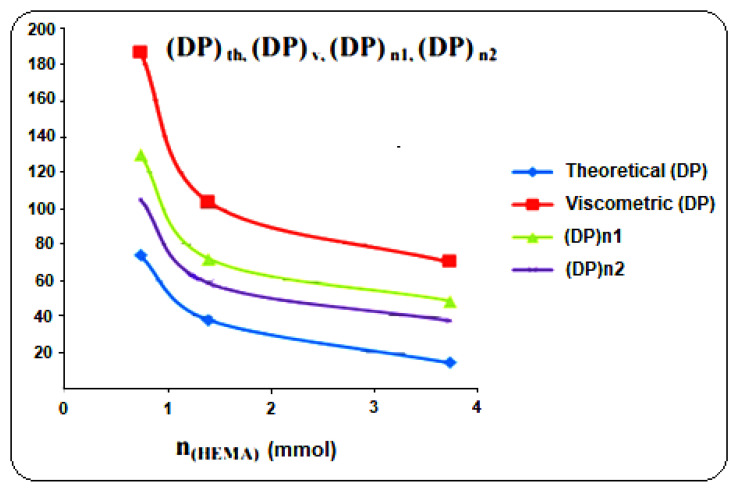
Evolution of degree of polymerization as a function of number of moles of initiator (HEMA) with DP. The theoretical degree of polymerization; (DP)v, viscometric degree of polymerization; (DP)n1, average degree of polymerization calculated by Equation (8); (DP)n2, average degree of polymerization determined from hydroxyl value.

**Figure 10 molecules-27-01408-f010:**
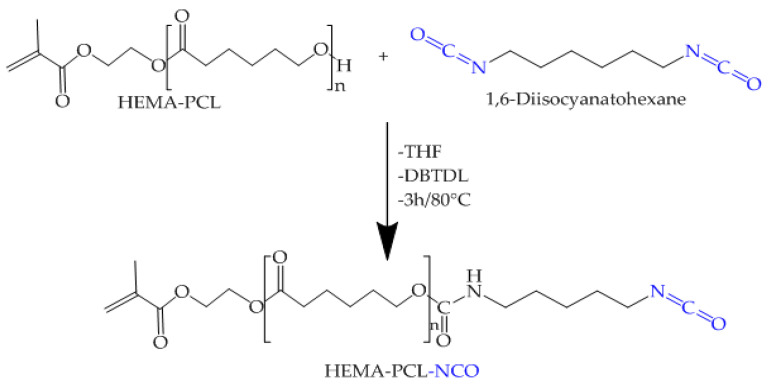
Synthesis of HEMA-PCL-NCO.

**Figure 11 molecules-27-01408-f011:**
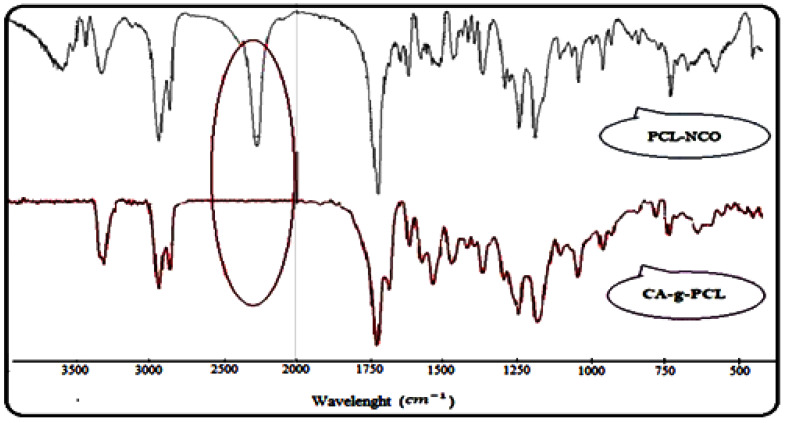
FTIR spectra of HEMA-PCL-NCO and CA-g-PCL copolymers.

**Figure 12 molecules-27-01408-f012:**
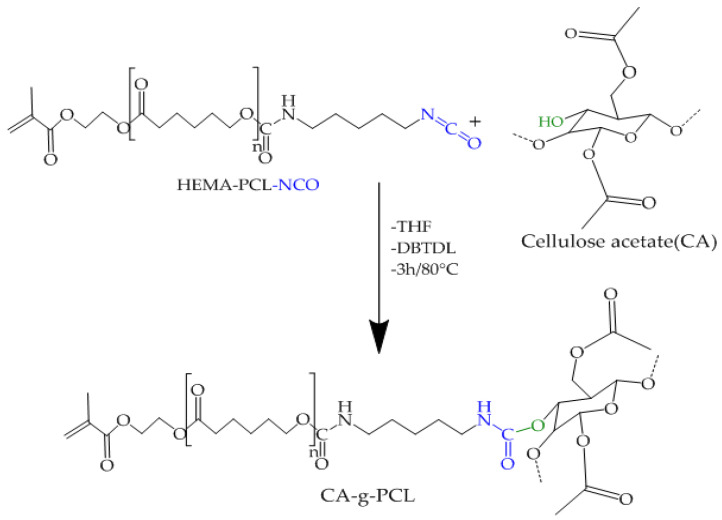
Synthesis of CA-g-PCL copolymers.

**Figure 13 molecules-27-01408-f013:**
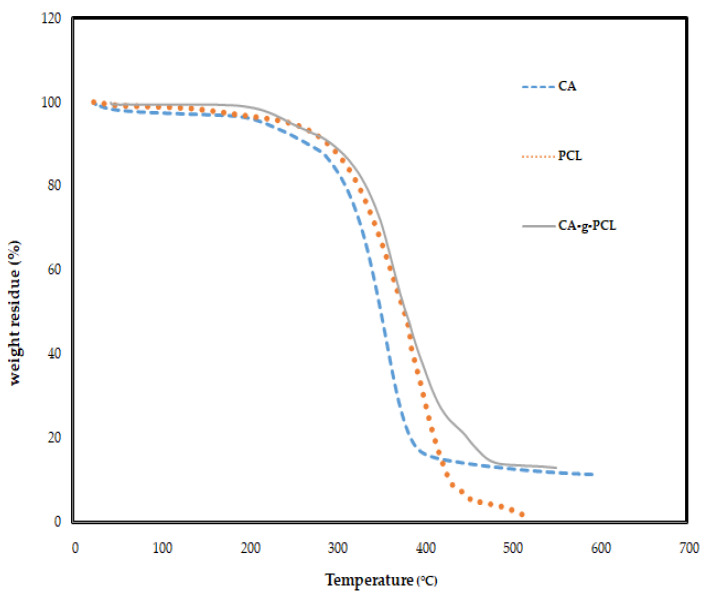
TGA thermograms of cellulose acetate, PCL, and CA-g-PCL copolymers.

**Figure 14 molecules-27-01408-f014:**
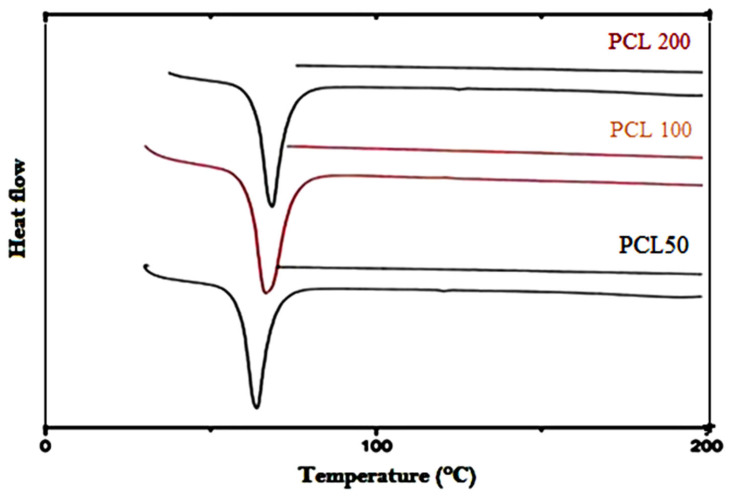
DSC thermograms of CA-g-PCL prepared with different DPs of PCL.

**Figure 15 molecules-27-01408-f015:**
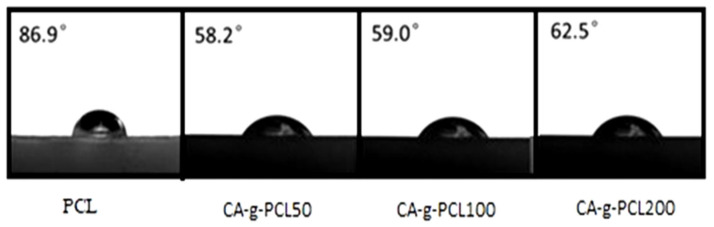
Contact angles for: PCL, CA-g-PCL50, CA-g-PCL100, and CA-g-PCL200.

**Figure 16 molecules-27-01408-f016:**
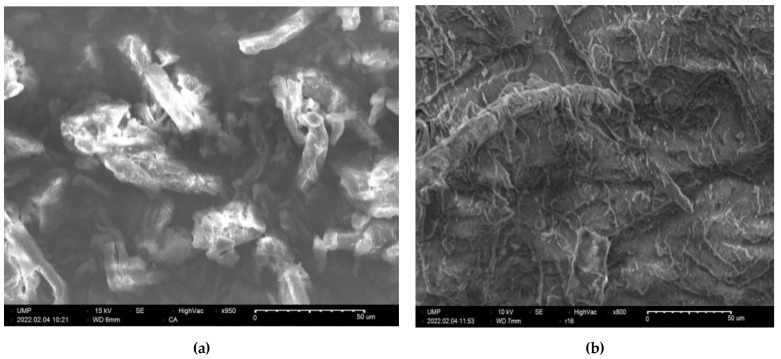
SEM image of (**a**) Cellulose acetate (**b**) CA-g-PCL.

**Figure 17 molecules-27-01408-f017:**
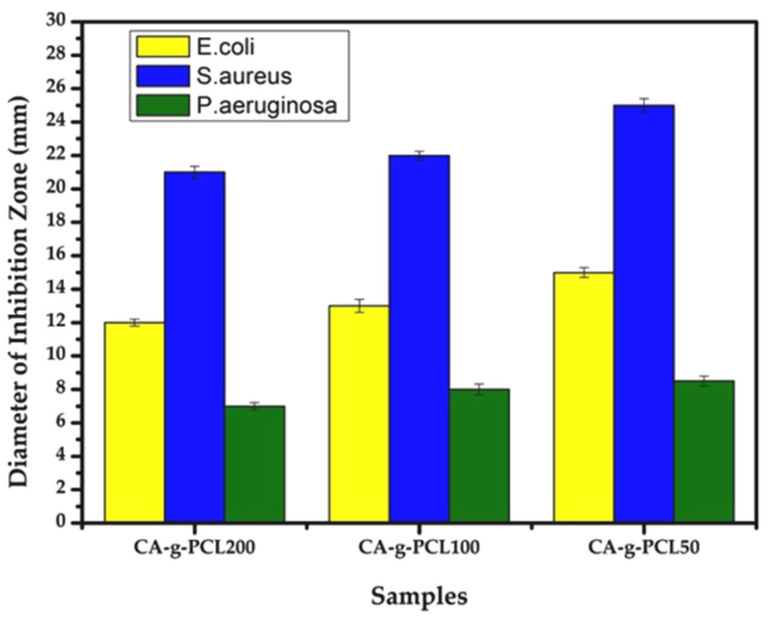
Antibacterial activity effect of CA-g-PCL50, CA-g-PCL100, and CA-g-PCL200.

**Figure 18 molecules-27-01408-f018:**
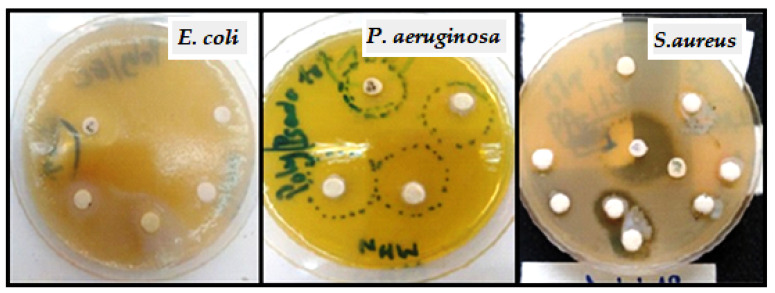
Antibacterial activity of CA-g-PCL50, CA-g-PCL100, and CA-g-PCL200 against three pathogenic bacteria: *E. coli*, *P. aeruginosa*, and *S.aureus* using the solid medium disk diffusion method.

**Figure 19 molecules-27-01408-f019:**
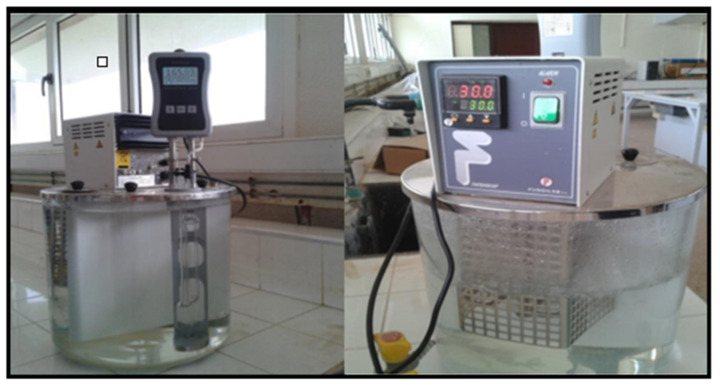
Picture of Cannon–Ubbelohdeviscometer for viscosity measurements.

**Table 1 molecules-27-01408-t001:** Flow time measurement results for HEMA-PCL50.

Sample 1	1 g/dL	1.5 g/dL	2 g/dL
T (s)	295.35	315.35	335.86
$\eta_{r}=\;t/t_{0}$	1.1428	1.2194	1.2987
$\eta_{\mathrm{sp}}={\;\eta }_{r}-1$	0.1428	0.2194	0.2987
$\eta_{\mathrm{red}}=\eta_{\mathrm{sp}}/C$	0.1428	0.1462	0.14935
$\eta_{\mathrm{inh}}=\;\mathrm{Ln}\left(\;\eta r\;\right)/C\;$	0.1335	0.13226	0.1307

**Table 2 molecules-27-01408-t002:** Flow time measurement results for HEMA-PCL100.

Sample 1	1 g/dL	1.5 g/dL	2 g/Dl
t (s)	308.42	335.44	365.2
$\eta_{r}=\;t/t_{0}$	1.1926	1.297	1.412
$\eta_{\mathrm{sp}}={\;\eta }_{r}-1$	0.1926	0.297	0.412
$\eta_{\mathrm{red}}=\eta_{\mathrm{sp}}/C$	0.1926	0.198	0.206
$\eta_{\mathrm{inh}}=\;\mathrm{Ln}\left(\;\eta r\;\right)/C\;$	0.17618	0.1733	0.1725

**Table 3 molecules-27-01408-t003:** Flow time measurement results for HEMA-PCL200.

Sample 1	1 g/dL	1.5 g/dL	2 g/dL
t (s)	334.65	374.32	415.52
$\eta_{r}=\;t/t_{0}$	1.294	1.4474	1.606
$\eta_{\mathrm{sp}}={\;\eta }_{r}-1$	0.294	0.4474	0.606
$\eta_{\mathrm{red}}=\eta_{\mathrm{sp}}/C$	0.294	0.2982	0.303
$\eta_{\mathrm{inh}}=\;\mathrm{Ln}\left(\;\eta r\;\right)/C\;$	0.2578	0.2465	0.236

**Table 4 molecules-27-01408-t004:** Results of intrinsic viscosity calculations with the Mark–Houwink–Sakurada equation and the equation of Naar et al.

Sample	[η] (dL/g) (±0.01 dL/g) (Mark–Houwink–Sakurada)	[η] (dL/g) (±0.01 dL/g) (Naar et al.)
HEMA-PCL50	0.137	0.1366
HEMA-PCL100	0.181	0.182
HEMA-PCL200	0.2785	0.267

**Table 5 molecules-27-01408-t005:** Results of calculation of the degree of polymerization and molar mass.

Sample	Conv(%) ^1^	[ε-CL]/[HEMA]	(DPn)Th ^2^	(DP)v	(DP)n	Mn(th) ^3^ (g/mol)	Mv ^4^ (g/mol)	Mn ^5^ (g/mol)
HEMA-PCL50	85.7	16.4	14.1	70	48.7	1734	8163	5574
HEMA-PCL100	87.14	43.6	38	103.6	72.1	4466.5	11956	8365
HEMA-PCL200	90	82	73.8	187	130	8553.5	21,575.5	15,057

^1^ The percentage of monomer conversion can be calculated as: Conv(%) = [(weight of reacted monomer)/(initial weight of monomer)] × 100; ^2^ Theoretical degree of polymerization DPn calculated as: (DP)n= [(n(monomer)/n (initiator)] × Conv(%); ^3^ Theoretical molecular weight Mn(th) was calculated as: Mn(th) = [(DP)n × M0] + extremity; ^4^ Viscosity mass can be calculated from the Equation (5): $\mathrm{Mv}={\left[\frac{\left[\eta \right]}{k}\right]}^{\frac{1}{a}}$; ^5^ Average molecular weight calculated as: Mn = DPn × M0, where Mo is the molecular weight of a repeating unit.

**Table 6 molecules-27-01408-t006:** Results of the hydroxyl value measurement for the three samples.

Sample	Blank (mL)	Sample (mL)	I(OH) (mg/gof Sample)	Mn(g/mol)	(DP)n
HEMA-PCL50	47.2	45.8	13.09	4285.7	37.5
HEMA-PCL100	47.2	46.3	8.415	6666.7	58.5
HEMA-PCL200	47.2	46.7	4.675	12,000	105

**Table 7 molecules-27-01408-t007:** Percent grafting and yield for CA-g-PCL copolymerization with different DPs.

Chain Length	Grafting (%)	Yield (%)
CA-g-PCL50	45	88
CA-g-PCL100	39	88
CA-g-PCL200	32	85

**Table 8 molecules-27-01408-t008:** Diameter of inhibition zone of samples against three bacteria.

Sample	Diameter of Inhibition Zone (mm)		
	*E. coli*	*S. aureus*	*P. aeruginosa*
CA-g-PCL200	12 ± 0.21mm	21 ± 0.35mm	7 ± 0.22mm
CA-g-PCL100	13 ± 0.39mm	22 ± 0.25mm	8 ± 0.33mm
CA-g-PCL50	15 ± 0.29mm	25 ± 0.4mm	8.5 ± 0.29mm

## Data Availability

Not applicable.

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
