# Peer review of "Cellulose Acetate-g-Polycaprolactone Copolymerization Using Diisocyanate Intermediates and the Effect of Polymer Chain Length on Surface, Thermal, and Antibacterial Properties"

_molecules, 2022, doi:10.3390/molecules27041408_

Round 1

Reviewer 1 Report

The manuscript “Cellulose acetate-g-polycaprolactone copolymerization using diisocyanate intermediates and the effect of polymer chain length on surface, thermal, and antibacterial properties” deals with the grafting of cellulose acetate with polycaprolactone by using diisocyanatedintermediates. The work is well organized and written. However, some revisions are required, as follows:

- Introduction. The state of the art related to the production of biopolymeric films with antimicrobial properties can be enlarged. For this purpose, see for instance this work: Baldino et al., Production, characterization and testing of antibacterial PVA membranes loaded with HA-Ag3PO4 nanoparticles, produced by SC-CO2 phase inversion, Journal of Chemical Technology and Biotechnology, 2019, 94(1), pp. 98–108; etc….

- Figure 4 is not readable. Quality of Figure 6, Figure 7 and Figure 8 is low.

- Results. Scanning electron microscopy analysis can be useful to observe the morphology of the obtained samples, as well as solvent residues analysis should be performed, since toxic organic solvents were used for sample preparation.

- Error bars in Figure 16 and Table 8 are missing.

Author Response

Dear Reviewer;

Thank you for your suggestions. please find in attached file our responses. all suggestions were treated.

Best regards

Reviewer 2 Report

  1. The authors didn’t mention the advantages of “grafting to” method and the reason they utilized it while this method has low efficiency compared to “grafting from” method as mentioned by the authors.
  2. The authors discussed about peaks of 1H NMR and FTIR analysis curves without any citation. Please add relevant citation(s) where possible. Also figures 4 and 5 have low qualities and should be replaced by high quality ones.
  3. The section for discussion of DP measurement by viscosity (section 2.2) is written in a very week format because: a) It is not known which solvent was used for dissolution of PCL and it is only found out after 2 pages that DMF is the solvent. b) Unit of measurement for viscosity has been not mentioned anywhere in the manuscript while it is necessary to check correctness of the presented results. c) Equations have not been numbered completely and some of the equations used for calculations don’t have any citation and have not been written in standard form. The equations should be written in separate lines with clear definition of all parameters. d) Values of parameters such as k, C, Mv, and a are not known. e) Figures 6 to 8 appeared without anu discussion in the text. e) Why a is 0.73 in eq. (1)? This section needs complete revision.
  4. If η is the symbol for viscosity, then the phrase “The recommended concentration range for good solvent viscosity measurement is: 1.1 ≤ η ≤ 1.4” is not correct and should be modified.
  5. It is recommended that all handwritten text throughout the manuscript such as first column of tables 1-3 and some figures be written by word processor or equation editor.
  6. Although the authors described some parameters used in the equations, but some parameters such as η, η0, c2, etc. are not introduced. Also, there is no Symbols section in the manuscript in order to know their definition. Please completely and precisely describe all parameters used throughout the manuscript.
  7. The quality of figures 6 to 16 is low and should be increased.
  8. In section 2.3, a definition was given for hydroxyl value without any citation. Also, the origin of the equations used in this section is not known. This section also needs more attention and modification.
  9. Please explain why the peaks for OH and NCO groups have been disappeared in CA-g-PCL spectrum.
  10. In first paragraph of section 2.6, two equations are embedded in the text which makes their reading and tracking hard. Also please explain the mentioned equation used for calculation of yield.
  11. It is recommended that the authors distinguish curves in Fig. 13 by using different symbols instead of colors.
  12. Finally, I recommend that the authors add recent works done in this field to the literature review section.

Author Response

(The authors gave the same response as above.)

Round 2

Reviewer 1 Report

The authors improved the manuscript.

Reviewer 2 Report

Accept after minor corrections.